# Epsins Negatively Regulate Aortic Endothelial Cell Function by Augmenting Inflammatory Signaling

**DOI:** 10.3390/cells10081918

**Published:** 2021-07-29

**Authors:** Yunzhou Dong, Beibei Wang, Kui Cui, Xiaofeng Cai, Sudarshan Bhattacharjee, Scott Wong, Douglas B. Cowan, Hong Chen

**Affiliations:** 1Vascular Biology Program, Boston Children’s Hospital, Harvard Medical School, Boston, MA 02115, USA; Yunzhou.Dong@childrens.harvard.edu (Y.D.); beibei.wang@childrens.harvard.edu (B.W.); kui.cui@childrens.harvard.edu (K.C.); sudarshan.bhattacharjee@childrens.harvard.edu (S.B.); scott.wong@childrens.harvard.edu (S.W.); douglas.cowan@childrens.harvard.edu (D.B.C.); 2Cardiovascular Biology Program, Oklahoma Medical Research Foundation, Oklahoma City, OK 73104, USA; xiaofeng-cai@omrf.org

**Keywords:** epsin, adaptor protein, endocytosis, adhesion molecule, selectin, MCP-1, endothelial activation, inflammation, TLR2/4, atherosclerosis, vascular disease

## Abstract

*Background:* The endothelial epsin 1 and 2 endocytic adaptor proteins play an important role in atherosclerosis by regulating the degradation of the calcium release channel inositol 1,4,5-trisphosphate receptor type 1 (IP3R1). In this study, we sought to identify additional targets responsible for epsin-mediated atherosclerotic endothelial cell activation and inflammation in vitro and in vivo. *Methods:* Atherosclerotic ApoE^−/−^ mice and ApoE^−/−^ mice with an endothelial cell-specific deletion of epsin 1 on a global epsin 2 knock-out background (EC-iDKO/ApoE^−/−^), and aortic endothelial cells isolated from these mice, were used to examine inflammatory signaling in the endothelium. *Results:* Inflammatory signaling was significantly abrogated by both acute (tumor necrosis factor-α (TNFα) or lipopolysaccharide (LPS)) and chronic (oxidized low-density lipoprotein (oxLDL)) stimuli in EC-iDKO/ApoE^−/−^ mice and murine aortic endothelial cells (MAECs) isolated from epsin-deficient animals when compared to ApoE^−/−^ controls. Mechanistically, the epsin ubiquitin interacting motif (UIM) bound to Toll-like receptors (TLR) 2 and 4 to potentiate inflammatory signaling and deletion of the epsin UIM mitigated this interaction. *Conclusions:* The epsin endocytic adaptor proteins potentiate endothelial cell activation in acute and chronic models of atherogenesis. These studies further implicate epsins as therapeutic targets for the treatment of inflammation of the endothelium associated with atherosclerosis.

## 1. Introduction

Atherosclerosis remains one of the most challenging cardiovascular diseases to treat and is the leading cause of heart attacks and strokes. Chronic inflammatory disease of the arteries underlies more than 50% of all deaths in westernized societies such as the United States [1]. Atherosclerosis involves multiple cell types, including endothelial cells (ECs), vascular smooth muscle cells (VSMCs), and immune cells such as mononuclear leukocytes [2,3]. One of the major pathological features of this disease is the accumulation of plaques (i.e., deposits of fat, cholesterol, calcium, and other substances) in the arterial wall [3]. Each of these cells can contribute to the progression of atherosclerosis; however, the layer of squamous endothelial cells that line the inner surface of blood vessels to form the endothelium are particularly important for initiation of atherosclerotic plaque formation [4,5,6,7].

Endothelial cell (EC) activation follows several well-established steps [8,9]. The first of which involves activation of innate inflammatory signaling pathways such as those initiated by the pattern-recognition Toll-like receptors TLR 2 and 4. As a result, the genetic deletion of TLR2, TLR4, or their downstream adaptor proteins (e.g., myeloid differentiation factor 88 (MyD88)) can reduce atherosclerosis in mice [10,11,12]. The second step is characterized by expression of adhesion molecules (e.g., ICAM-1, VCAM-1, P-selectin, and E-selectin) and chemoattractant proteins (e.g., MCP-1) on the luminal EC surface, which encourages docking of immune cells (e.g., monocytes, neutrophils, and leukocytes) [13,14,15]. Lastly, the immune cells migrate into the intima and differentiate into macrophages that engulf substances such as oxidized low density lipoprotein (oxLDL) to become foam cells [16]. In this arterial injury model, increased circulating cholesterol, oxLDL, and cytokines activate inflammatory signaling [14,17,18] to promote the expression of adhesion and chemoattractant proteins in regions with disturbed blood flow (e.g., vessel branch points) [19]. The result is immune cell infiltration, foam cell formation, and deposition of excessive cholesterol in the arterial intima [20].

As receptor-mediated inflammatory signaling is central to the initiation and progression of atherosclerosis, and the uptake and sorting of EC surface receptors can fine-tune these signaling pathways [21,22,23,24,25,26,27], we focused our studies on the epsin family of clathrin-dependent endocytic adaptors [28,29,30]. In particular, epsins 1 and 2 modulate vascular endothelial growth factor receptor 2 (VEGFR2) signaling in ECs [21,23,24,26]. Using epsin mutant mouse models, we discovered that epsins are also involved in regulating Notch, VEGFR3, and Wnt signaling pathways [16,21,22,23,24,26,31]. By binding to ubiquitinated membrane receptors, epsins control physiological and pathological regulation of embryogenesis [16], angiogenesis [21,23,24,26], lymphangiogenesis [22] and cancer progression [25,31]. In addition, we recently reported that epsins 1 and 2 fuel atherosclerotic degradation of the calcium release channel inositol 1,4,5-trisphosphate receptor type 1 (IP3R1) [32].

The latter studies indicate that other mechanisms were also involved in the pathogenesis of atherosclerosis because of epsin-deficiency. In this study, we further assessed the role of epsins in endothelial activation and inflammatory signaling in atherogenesis. Using biochemical, molecular, and genetic approaches, we demonstrated that epsins 1 and 2 are required for endothelial activation of inflammatory signaling pathways, and that the loss of epsins from the endothelium reduced expression of ICAM-1, VCAM-1, P-selectin, and E-selectin adhesion molecules as well as the chemoattractant protein MCP-1. Loss of endothelial epsins blunted inflammatory signaling in murine aortic endothelial cells (MAECs) in vitro and reduced atherosclerosis in vivo. Together, our data reveal epsins to be potentially valuable therapeutic targets for the treatment of atherosclerosis and other diseases.

## 2. Materials and Methods

### 2.1. Mouse Strains and Models

Mouse strains and procedures were approved by the Institutional Animal Care and Use Committee (IACUC) at Boston Children’s Hospital and the Oklahoma Medical Research Foundation. Male and female mice were used and housed using a 12/12 h light/dark cycle. The knockout of both epsin 1 and 2 causes embryonic lethality [16]. To avoid this, we generated an epsin 1 loxP mouse strain by inserting *loxP* sites flanking exon 2 in an epsin 2 null background (epsin 1^fl/fl^:epsin 2^−/−^) to create a conditional deletion of epsin 1 postnatally [21,23]. Global epsin 1/2 knockout mice were produced by crossing epsin 1^fl/fl^:epsin 2^−/−^ mice with a β-actin promoter-driven Cre transgenic mouse strain [32]. The endothelial cell (EC)-specific deletion of epsin mouse strain was established by crossing epsin 1^fl/fl^: epsin 2^−/−^ mice with vascular endothelial cell-specific Cre transgenic mice (VECad-Cre) [23], which were bred with atherosclerotic apolipoprotein E null mice (ApoE^−/−^) (Jackson Laboratory) and backcrossed seven times. Mice were treated with 4-hydroxytamoxifen (5–10 mg/kg, body weight) 5 to 7 times every other day at 8 weeks of age [33]. Atherosclerosis was accelerated in these mice by feeding them an atherogenic diet (i.e., Western Diet (WD)) that contained 1.3% cholesterol and 0.5% cholic acid (TD 02028; Harlan Teklad). Mice were fed a WD starting at 8 weeks of age for 12 to 14 weeks [34].

### 2.2. Cell Culture Models

Murine aortic endothelial cells (MAECs) were harvested and cultured as described previously [32]. Using a dissecting microscope, aortas were collected from mice and fat and connective tissue were carefully removed [32]. Aortas were opened longitudinally, cut into 2 to 3 segments, and placed in a 6-well plate that was coated with Matrigel (BD Biosciences). After 4 days, EC networks were visible by brightfield microscopy. Aortic segments were discarded, and ECs were cultured for 2 more days. MAECs were detached with Dispase II (Roche, Mannheim, Germany) and cultured in fresh EC medium. The identity of MAECs was confirmed by immunofluorescence staining with CD31, von Willebrand factor, α-smooth muscle actin, and VEGFR2 antibodies [32]. The purity of isolated MAECs was determined by cytometry using a VE-cadherin antibody [35]. We typically obtain approximately 90% purity. Freshly isolated MAECs were treated with 5 µM tamoxifen for 4 days to induce the deletion of the epsin 1 gene from epsin 1^fl/fl^:epsin 2^−/−^:β-actin-Cre, epsin 1^fl/fl^:epsin 2^−/−^:VECad-Cre, or iCDH5-Cre mice [32]. Epsin 1 and 2 protein levels were monitored by Western blot (Appendix A). MAECs were also treated with oxLDL and 7-ketocholesterol (7-KC) as previously described [32].

### 2.3. Evaluation of Atherosclerosis

Oil Red O (ORO) staining of aortic roots and arches was performed as previously described [32,34,36]. In brief, mouse hearts were fixed in 4% paraformaldehyde (PFA) in phosphate buffered saline (PBS) for 16 h and then embedded in OCT tissue freezing compound. Eight aortic arch sections were collected from each mouse and stained with 0.5% ORO, and counterstained with hematoxylin. Atherosclerotic plaques were imaged with an Olympus microscope. For *en face* aortic arch staining, the intimal surface was exposed by a longitudinal cut, laid flat, and fixed in 10% formalin overnight [37]. Arches were washed with dH_2_O three times and air dried for 10 min. Following incubation in 100% propylene glycol for 2 to 5 min, tissues were stained at 60 °C in 0.5% ORO solution for 8–10 min at 60 °C. Tissues were then differentiated in 85% propylene glycol for 2 to 5 min, and rinsed twice with dH_2_O. Digital images of the aortic arches were captured using a stereomicroscope, and the lesional area was quantified using Image J (available from the National Institutes of Health (NIH) website: https://imagej.nih.gov/ij/ (accessed on 30 March 2012).

### 2.4. Protein Binding Analyses

MAECs treated with 1 µg/mL LPS [38] for the indicated times were lysed in RIPA buffer containing 50 mM Tris-HCl (pH 7.4), 150 mM NaCl, 5 mM EDTA (pH 8.0), 30 mM NaF, 1 mM Na_3_VO_4_, 40 mM β-glycerophosphate, 1x protease inhibitors (Roche), 20 mM N-Ethylmaleimide, 10% glycerol, and either 1% Nonidet-P40 or Triton X-100. One mg of protein was used for immunoprecipitation (IP). Primary antibodies were added to the protein lysates and gently agitated for 4 to 16 h. Then, 35 to 40 µL rec-G beads (Life Technologies, Carslbad, CA, USA) were added for another 4 h. Beads were washed with IP buffer containing protease inhibitors and N-Ethylmaleimide five times. Samples were mixed with 2× loading buffer and incubated at 95 °C for 5 min before Western blot analysis [32].

### 2.5. Standard Experimental Procedures

Molecular cloning, Western blotting, histological staining, immunofluorescence staining, brightfield and fluorescence microscopy, and cell culture were performed according to standard procedures [21,23,26,31,32,36,39]. DNA or siRNA transfection, electroporation, RT-PCR, FACS analyses and molecular cloning were performed as previously described [21,23,26].

### 2.6. Statistical Analyses

Statistical analyses were conducted using Prism 8.0 (Graph Pad Software, San Diego, CA, USA) [32]. Data are presented as mean ± standard error of the means (SEM). Data were analyzed by two-tailed, unpaired or paired Student’s *t*-test or ANOVA with Bonferroni’s procedure was used for multiple comparisons. A *p*-value of less than 0.05 was considered statistically significant.

## 3. Results

### 3.1. Epsins Augment Inflammatory Signaling in Atherosclerotic Endothelial Cells

#### 3.1.1. Epsin Loss Attenuates Inflammatory Signaling in Activated Endothelial Cells

Due to embryonic lethality in global epsin 1 and 2 knockout mice [16], we established a conditional deletion of epsin 1 on an epsin 2-null background to enable more exact analyses of the function of these redundant endocytic adaptor proteins [23]. This strain was crossed with β-actin promoter-driven Cre and vascular endothelial specific-cadherin (VECad) or inducible cadherin-5 (iCDH5) promoter-driven Cre (Appendix A) transgenic mice to isolate primary cultured murine aortic endothelial cells (MAECs). An atherosclerotic epsin double knockout mouse strain was then generated by breeding epsin iDKO onto an ApoE^−/−^ background (Appendix A). Epsin 1 and 2 deletion from isolated MAECs was carried out in vitro by adding 5 µM tamoxifen for 4 days and confirming epsin deletions by Western blot analyses (Appendix A).

As the expression of adhesion molecules, cytokines, and chemoattractant proteins are controlled by inflammatory signaling, we analyzed TNF-α and LPS stimulated signaling in MAECs. TNF-α and LPS treatments attenuated phospho-NF-κB (p65) levels in EC-iDKO MAECs (Figure 1A,B and Appendix A). At the same time, MAPK pathways including phospho-p38 and phospho-JNK were also decreased in cells lacking epsins (Figure 1C,D; Appendix A). Immunofluorescence staining showed that epsin deficiency reduced nuclear translocation of p65 from the cytoplasm of MAECs (Figure 1E). These data suggest that epsin loss in ECs inhibits TNF-α and LPS-mediated inflammatory signaling.

#### 3.1.2. Epsin Loss Inhibits Endoplasmic Reticulum Stress in Atherosclerosis

Endoplasmic reticulum stress (ER stress) is an important consequence of cardiovascular inflammation [40]. As observed in human atherosclerosis patients, the ER stress markers KDEL and XBP-1 are increased compared with unaffected controls (Appendix A) [32]. In our in vivo models, we showed that ER stress is significantly diminished in EC-iDKO/ApoE^−/−^ mice compared to ApoE^−/−^ controls (Figure 2A,B) by measuring KDEL, ATF6, and XBP-1 in the endothelium using immunofluorescence co-staining with the endothelial cell marker CD31.

To complement the above observation mechanistically, we isolated MAECs from ApoE^−/−^ and EC-iDKO/ApoE^−/−^ mice, and then treated the cells with atherogenic oxLDL. Epsin-deficient MAECs significantly attenuated oxLDL-induced ER stress as exhibited by the reduced ER stress markers P-PERK, P-eIF2α, P-JNK, and ATF6 (Figure 2C,D), which is further confirmed by 7-KC (7-Ketocholesterol) treatment in MAECs isolated from ApoE^−/−^ and EC-iDKO/ApoE^−/−^ mice (Figure 2E,F; Appendix A).

#### 3.1.3. Epsin Loss Reduces Expression of Endothelial Adhesion Molecules In Vitro

To investigate the role of epsins in endothelium activation, we measured the expression of adhesion molecules (including ICAM-1, VCAM-1, P-selectin, and E-selectin) and chemoattractant protein (MCP-1) in cultured primary MAECs isolated from wild type (WT) or epsin-deficient mice (EC-iDKO) by flow cytometry and quantitative RT-PCR (Figure 3). The plasma membrane levels of ICAM-1 and VCAM-1 were considerably reduced as determined by flow cytometry (Figure 3A–C). Similarly, the expression of MCP-1 was significantly reduced in the DKO MAECs stimulated by LPS (Figure 3D,E).

As the expression of these molecules is largely regulated at the transcriptional level, we also performed unsaturated RT-PCR for 20 cycles (the primers are listed in Appendix A), followed by 1% agarose gel electrophoresis. Our results demonstrate that the expression of adhesion molecules and MCP-1 in MAECs lacking epsins 1 and 2 are significantly reduced when compared to control (WT) cells (Figure 3F). We confirmed these findings using by real-time qRT-PCR (Figure 3G–J), which showed the loss of epsins in mouse ECs significantly attenuated expression of adhesion molecules (ICAM-1, VCAM-1) and selectins (P-selectin, and E-selectin). Together, these data support the conclusion that endothelial epsins are required for the endothelial activation.

#### 3.1.4. Epsin Loss Reduces Endothelial Adhesion Molecule Expression In Vivo

We measured ICAM-1, VCAM-1, and P-selectin expression in ApoE^−/−^ control and EC-iDKO/ApoE^−/−^ mice fed a Western diet (WD) for 8 weeks. Immunohistochemical (IHC) staining of frozen heart sections revealed that ICAM-1, VCAM-1 and P-selectin expression in epsin-deficient endothelium was significantly reduced (Figure 4A,B). These results were verified by immunofluorescence staining (Figure 4C,D), which showed greater macrophage/monocyte accumulation (Moma-2) and P-selectin staining in the ApoE^−/−^ endothelium, when compared with the same tissue in EC-iDKO/ApoE^−/−^ mice. These data indicate that epsins are required for endothelium activation in vivo.

#### 3.1.5. Loss of Endothelial Epsins Attenuates Atherosclerosis and Oxidative Stress

As shown above, epsins are crucial for early inflammatory responses in activated endothelium, as assessed by expression analyses of adhesion molecules and MCP-1. As a result, we expect to observe a reduction in atherosclerotic plaques in EC-iDKO/ApoE^−/−^ mice fed a WD (Appendix A). Genetic loss of epsins in the endothelium significantly attenuated atherosclerosis as evidenced by the reduced number and size of plaques in both aortic roots and arches as shown using ORO staining (Figure 5A–C).

In support of our observations in EC-iDKO/ApoE^−/−^ mice, we assessed oxidative stress markers (3-nitrotryrosin (3-NT) and oxLDL) by immunofluorescence staining. 3-NT and oxLDL were both reduced in epsin-deficient mice (Figure 5D–G). These data demonstrate that epsin loss from the endothelium reduces atherogenesis and oxidative stress—an important hallmark of inflammation.

#### 3.1.6. Epsins Bind Toll-like Receptors 2 and 4 to Potentiate Inflammatory Signaling

To further understand the role of epsins in inflammatory signaling, we treated MAECs with LPS for 0, 15, and 30 min. LPS treatment induced binding interactions between epsin 1, TLR2, TLR4, and MyD88 (Figure 6A; Appendix A). Interestingly, there was no change in the expression of TLR2 and TLR4 because of LPS treatment using Western blotting and low cytometry (Appendix A). In a reciprocal IP with TLR4, epsin 1 and MyD88 were found to bind TLR4 (Figure 6B; Appendix A). To confirm these results, we used MyD88 for IP and Western blot detection of TLR2/4 and epsin 1, which were also detected (Figure 6C; Appendix A). In LPS signaling, inflammatory mediators such as NF-κB can interact with receptor interacting protein 1 (RIP1) to induce cell death. Additional IP analyses showed that epsin 1 interacts with this protein (Figure 6D; Appendix A).

As MAPK signaling pathways (JNK and p38) are also blocked by epsin deletions (Figure 1), we speculated that TRAF6, a downstream kinase for MAPK signaling, may also bind epsins. To test this, we performed IP experiments to show that epsin 1 interacts with TRAF6 under inflammatory stimulation (Figure 6E; Appendix A). As the epsin ubiquitin interacting motif (UIM) has been proven to be critical for activated receptor degradation [23,33], we proposed that this domain may be important for the epsin-TLR2/4 interaction. Using deletion constructs, we found the interaction between epsin 1 and TLR2/4 was largely abolished (Figure 6F; Appendix A) and the same was true for the epsin 1/MyD88 interaction (Figure 6G; Appendix A).

Lastly, we measured blood cytokine levels in our mouse models, as these molecules are the effectors of inflammation. LPS stimulated release of INF-γ, IL-6, and TNF-α into the blood was significantly reduced in EC-iDKO/ApoE^−/−^ mice compared to WT/ApoE^−/−^ mice (Figure 6H), which strongly suggests that epsin loss from the endothelium reduces cytokine production under inflammatory conditions (e.g., LPS stimulation). Taken together, our results indicate that epsins interact with the TLR2/4 machinery to promote inflammatory signaling through modulation of adhesion molecules, chemoattractants, and cytokines (Figure 6I).

#### 3.1.7. Epsin Loss Has no Effect on Glucose or Lipid Levels in Atherosclerotic Mice

It is worth noting that glucose and lipid profiles of ApoE^−/−^ and EC-iDKO/ApoE^−/−^ mice were not changed. This implies that epsin deficiency does not systemically affect whole body glucose or lipid metabolism (Appendix A).

## 4. Discussion

Here we show a novel role for the epsin 1 and 2 endocytic adaptor proteins in endothelial cell (EC) activation and atherogenesis. Using epsin mutant mice and in vitro approaches, we demonstrate that epsins are important regulators of inflammatory signaling pathways and downstream effectors including adhesion molecules and chemokines. This study links epsins to endothelium activation in response to inflammatory stimuli prevalent in vascular diseases such as atherosclerosis.

Endothelial activation is one of the key mechanisms of inflammation that initiates atherosclerosis [7,14,41]; however, the underlying molecular mechanisms remains elusive. In the widely accepted ‘endothelium injury’ model, EC activation plays a critical role in atherosclerosis [4,7,42]. The events that occur in the early stages of this disease include EC activation followed by immune cell-endothelial cell interactions [43], resulting from the expression of molecules such as ICAM-1, VCAM-1, P-selectin, and MCP-1 on the EC surface [18,44,45,46]. Under inflammatory conditions, such as exposure to oxidized low density lipoprotein (oxLDL) and circulating cytokines [47,48], the expression of adhesion molecules attracts immune cells that adhere to and infiltrate the endothelium through the process of tethering, adhesion, rolling, and migration [13].

Once in the intima, leucocytes and monocytes differentiate into macrophages that engulf oxLDL to form foam cells. Our results show that loss of epsins in ECs attenuates activation of the endothelium and reduces expression of adhesion molecules and MCP-1. In endothelial epsin-deficient mice, oxidant stress and ER stress are both significantly abrogated (Figure 2 and Figure 5), which are critical components of inflammation [49]. From a clinical perspective, the CANTOS trial clearly establishes that a reduction in inflammation in patients using Interleukin-1β antibodies (i.e., canakinumab) significantly reduces the risk of atherosclerotic disease in humans [50].

Using endothelial-specific epsin-deficient mice, we found that these proteins play a critical role in mediating inflammatory signaling through the modulation of the NF-κB and MAPK pathways (i.e., the JNK and p38 pathways) (Figure 1 and Figure 2). These signaling cascades are important for the expression of adhesion molecules, cytokines, and chemokines. In particular, the loss of epsins attenuates expression of these molecules in in vitro and in vivo models of atherosclerotic inflammation (Figure 3 and Figure 4).

Endothelial dysfunction can be induced by multiple mediators, such as those found in diabetes, hypertension, obesity, inflammation, and smoking. These pathological conditions elevate the generation of reactive oxygen species (ROS) and reactive nitrogen species (RNS) and are characterized by an imbalance in intracellular calcium handling and ER stress [32], among other mediators, which includes free fatty acids and advanced glycation end products, insulin resistance, and hyperlipidemia, apoptosis, hyperinsulinemia, and hyperglycemia [4]. These insults eventually affect the function of the endothelium, which can initiate atherogenesis. The loss of endothelial epsins can improve endothelial function in blood vessels and reduce inflammation. Our data suggest that epsin loss is beneficial by maintaining EC homeostasis under inflammatory conditions.

## 5. Conclusions

We demonstrated that the loss of epsins 1 and 2 in endothelial cells inhibits endothelium activation during acute and chronic inflammation through reduced expression of adhesion molecules and the chemoattractant protein MCP-1. Mechanistically, epsins, at least in part, modulate inflammatory signaling to potentiate endothelial activation by interacting with components of the Toll-like receptor signaling pathway (Figure 6). Our studies suggest that epsins represent therapeutic targets for the treatment of inflammation of the endothelium associated with atherosclerosis (Figure 7).

## 6. Patents

U.S. patent application #20120197059: Motif mimicry of epsins in cancer and metabolic therapeutics. Inventors: H.C. and Y.D.

## Figures and Tables

**Figure 1 cells-10-01918-f001:**
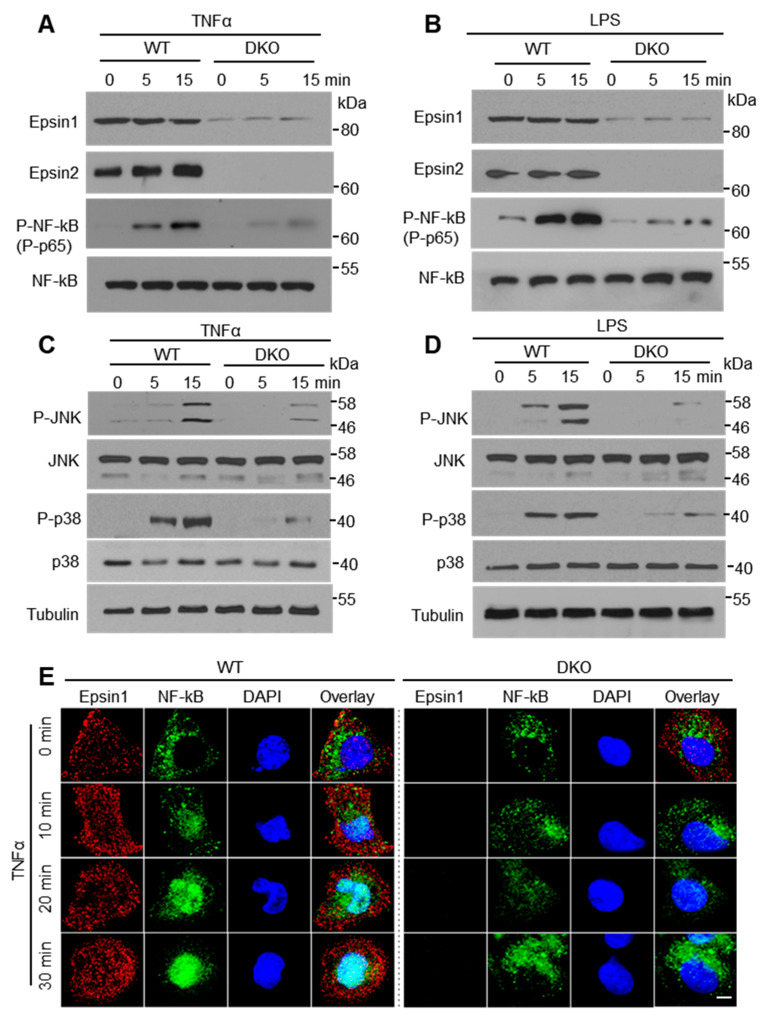
Loss of epsins in MAECs attenuates inflammatory signaling. (**A**,**B**) Western blot analyses of NF-κB (p65) signaling induced by TNF-α (50 ng/mL) or LPS (1 µg/mL) (n = 4). (**C**,**D**) Western blot analyses of MAPK (p38 and JNK) signaling induced by TNF-α (50 ng/mL) or LPS (1 µg/mL) (n = 4). (**E**) NF-κB (p65) translocation in WT or DKO MAECs stimulated with 50 ng/mL TNF-α for the indicated time (n = 4). Scale bar = 10 µm.

**Figure 2 cells-10-01918-f002:**
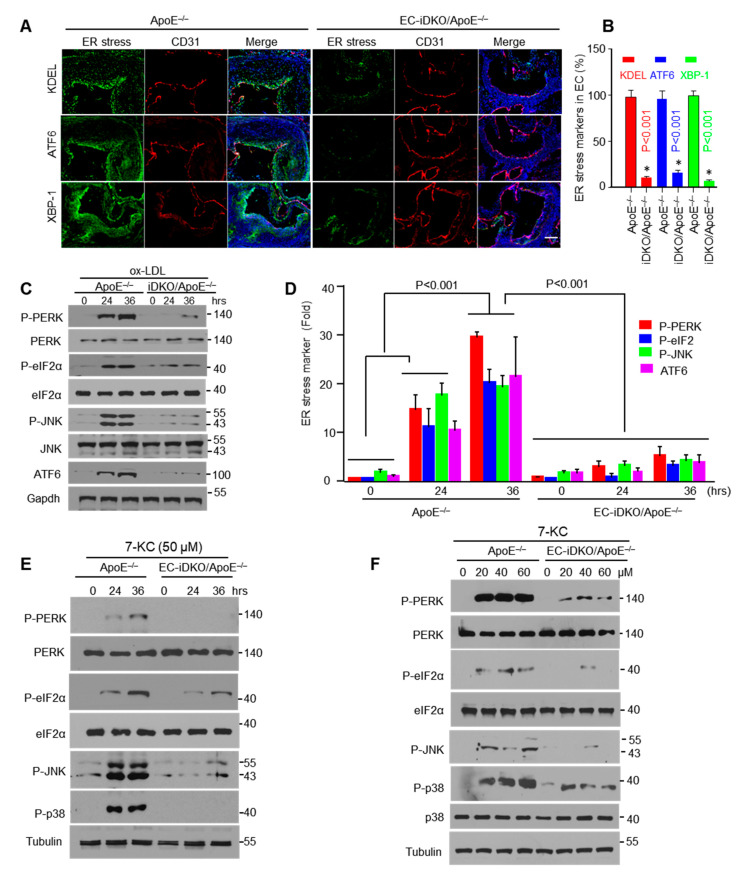
Loss of endothelial epsins attenuates ER stress-induced inflammatory signaling. (**A**) Comparison of endothelium ER stress markers in the aortic roots of ApoE^−/−^ or EC-iDKO/ApoE^−/−^ mice by immunofluorescence staining. CD31 served as an endothelium marker (n = 5 in each group). Scale bar = 500 µm. (**B**) Quantification of ER stress markers from A. * ApoE^−/−^ vs. EC-iDKO/ApoE^−/−^. (**C**) MAECs were isolated from ApoE^−/−^ or EC-iDKO/ApoE^−/−^ mice, followed by treatment with ox-LDL at 100 µg/mL for the indicated times. Cell lysates were subjected to Western blot analyses using the specified antibodies. (**D**) Quantification of results from C (n = 5). (**E**,**F**) MAECs with or without epsins were treated with 7-KC for various times and concentrations as indicated, and ER stress markers were assessed by Western blot analysis (n = 3).

**Figure 3 cells-10-01918-f003:**
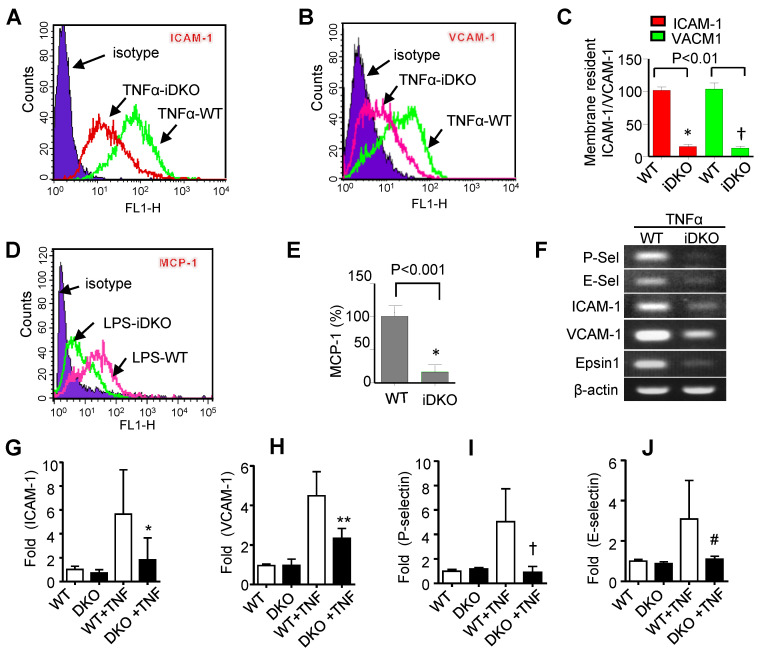
Loss of endothelial epsins impairs expression of inflammatory genes. (**A**,**B**) Cell surface presentation of adhesion molecules (ICAM-1 and VCAM-1) as determined by flow cytometry after treatment with 50 ng/mL TNF-α for 16 h (n = 3). (**C**) Statistical histogram for B and C (* or †, *p* < 0.01). (**D**,**E**) Cell surface presentation of MCP-1 by flow cytometry after stimulation with 1 µg/mL LPS for 3 h (n = 3). (**E**) Quantification for (**D**). (**F**) Expression of adhesion molecules in WT and DKO MAECs after 50 ng/mL TNF-α stimulation for 3 h analyzed as determined by unsaturated PCR analysis and agarose gel electrophoresis. (**G**–**J**) Analysis of adhesion molecules (ICAM-1 and VCAM-1) and selectins (P- and E-selectin) in MAECs stimulated with TNFα (50 ng/mL) for 3 h, followed by extraction of total RNA, cDNA synthesis, and real-time qRT-PCR (n = 3–5; *, **, †, or #, *p* < 0.05).

**Figure 4 cells-10-01918-f004:**
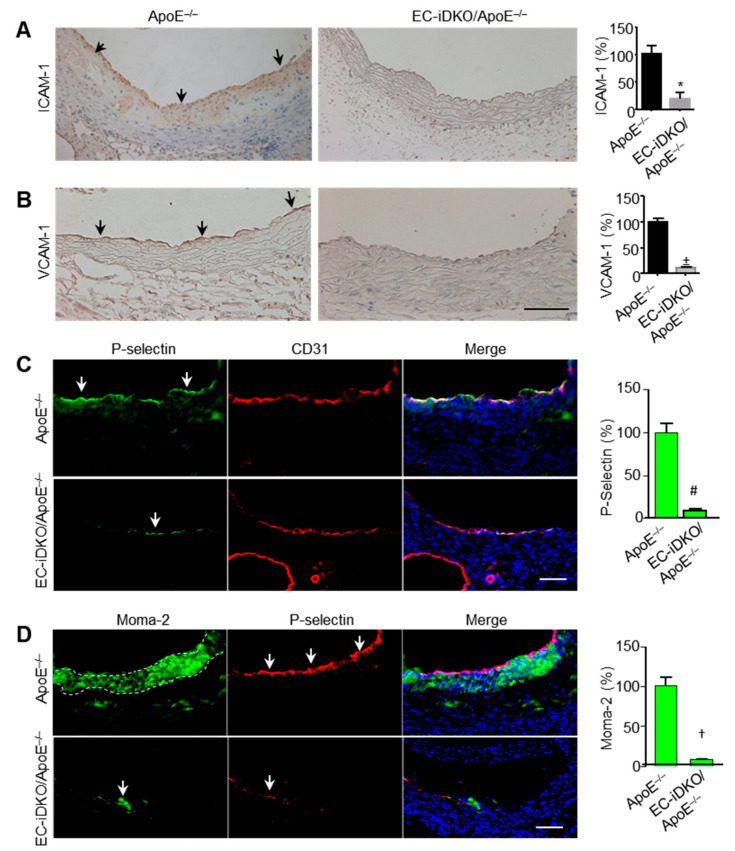
Epsin loss attenuates adhesion molecules expression in ApoE-null mice. (**A**,**B**) ICAM-1 and VCAM-1 expression in EC-iDKO/ApoE^−/−^ and ApoE^−/−^ mice fed a WD for 8 weeks as determined by immunohistochemical (IHC) staining (n = 5 in each group; *, +, ApoE^−/−^ vs. EC-iDKO/ApoE^−/−^; *p* < 0.001). Black arrows indicate ICAM-1 or VCAM-1. Scale bar = 100 µm. (**C**) P-selectin and CD31 staining in the endothelium of aortic roots from ApoE^−/−^ and EC-iDKO/ApoE^−/−^ mice fed a WD for 8 weeks (n = 5 mice in each group; # *p* < 0.01). White arrows indicate P-selectin. Scale bar = 100 µm. (**D**) Moma-2 and P-selectin staining in ApoE^−/−^ and EC-iDKO/ApoE^−/−^ aortic roots (n = 5 mice in each group; † *p* < 0.01). White arrows indicate P-selectin. The dashed line indicates macrophage/monocyte accumulation (Moma-2). Scale bar = 100 µm.

**Figure 5 cells-10-01918-f005:**
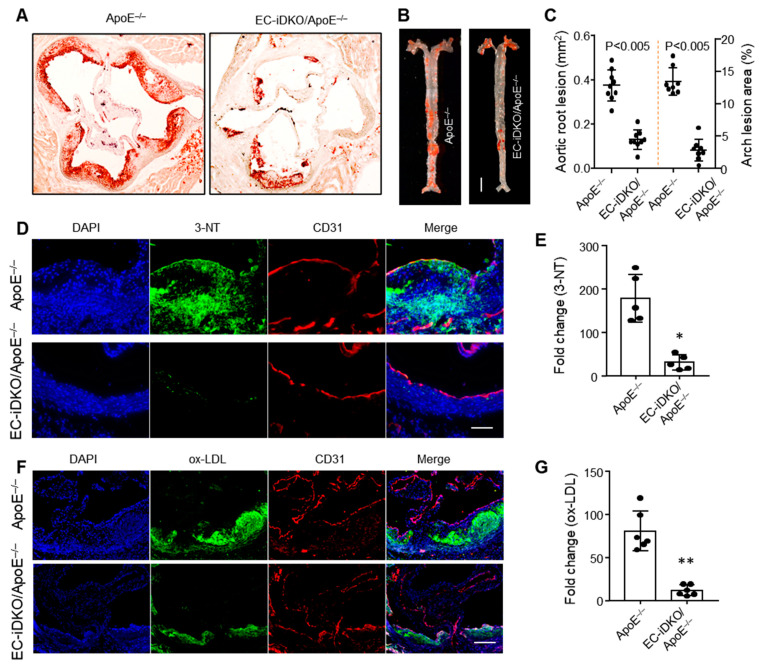
Loss of epsins from the endothelium mitigates atherosclerosis and oxidative stress. (**A**–**C**) Oil Red O staining for aortic roots, aortic arches, and data quantification, respectively (n = 9 in each group; *p* < 0.001). Scale bar = 500 µm (**A**) or 5 mm (**B**). (**D**,**E**) Immunofluorescence staining of 3-NT for reactive nitrogen species (RNS) and quantification (n = 5; * ApoE^−/−^ vs. EC-iDKO/ApoE^−/−^; *p* < 0.005). (**F**,**G**) Immunofluorescence staining of oxLDL for reactive oxygen species (ROS) and quantification (n = 5; ** ApoE^−/−^ vs. EC-iDKO/ApoE^−/−^; *p* < 0.005).

**Figure 6 cells-10-01918-f006:**
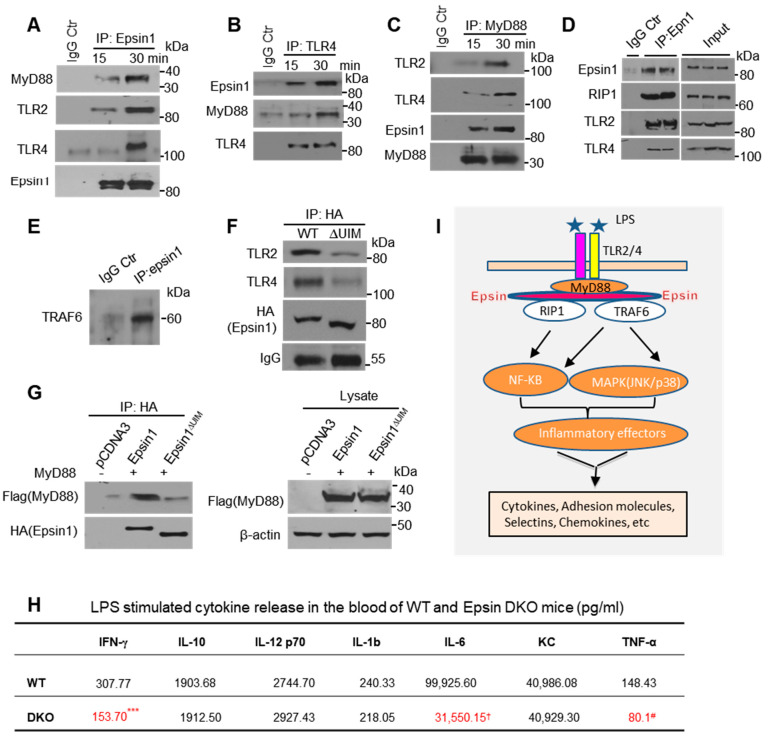
Epsins interact with the TLR2/4 signaling complex to exacerbate inflammation. (**A**) Wild type MAECs were treated with 100 ng/mL LPS for 15 or 30 min and lysates were used for IP with an epsin 1 antibody prior to Western blotting with MyD88, TLR2, and TLR4 antibodies (n = 5). (**B**) The same treatment as described above using a TLR4 antibody for IP and epsin 1, MyD88, and TLR2/4 antibodies for Western blotting (n = 5). (**C**) The same treatment as described above using MyD88 antibody for IP and blotted with epsin 1, TLR2, and TLR4 antibodies (n = 5). (**D**) LPS-treated MAECs (30 min) were used for IP with the epsin 1 antibody and blotting with RIP1, TLR2, and TLR4 antibodies (n = 5). (**E**) LPS-treated MAECs were used for IP with an epsin 1 antibody and blotted with a TRAF6 antibody (n = 5). (**F**) Epsin 1 wt (HA tag) or epsin 1ΔUIM (HA tag) constructs were transfected to WT MAECs by electroporation. After 30 h, IP experiments using a HA antibody were performed and blotted using HA, TLR2, and TLR4 antibodies (n = 3). (**G**) Epsin 1 wt (HA tag) or epsin 1ΔUIM (HA tag) were co-transfected to MAECs with MyD88 (Flag tag) by electroporation. After 30 h, cells were lysed and IP was performed with a HA antibody, followed by blotting with Flag or HA antibodies. (**H**) Cytokines between ApoE^−/−^ and EC-iDKO/ApoE^−/−^ mice stimulated by LPS (10 mg/kg, IP injection) after 4 h. Serum was collected and cytokines were measured by ELISA (RND Systems). ***, INFγ; †, IL-6; #, TNFα; WT vs. DKO mice; all *p* < 0.05. (**I**) Diagram showing how epsins modulate inflammatory signaling by potentiating NF-κB and MAPK signaling.

**Figure 7 cells-10-01918-f007:**
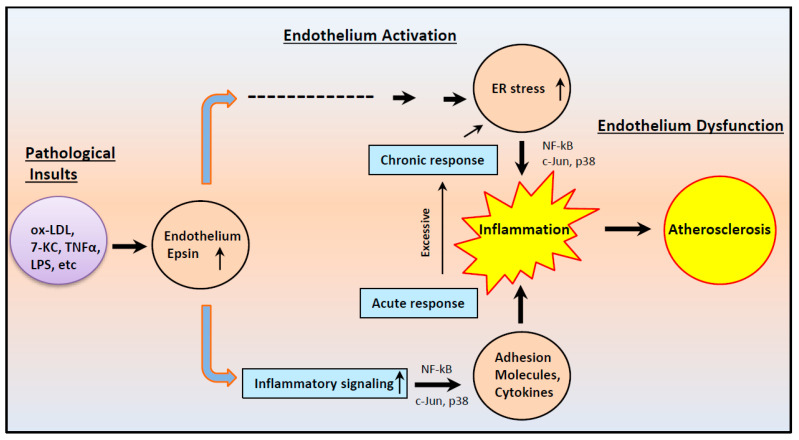
Schematic diagram of the role of epsins in mediating atherosclerosis. Under pathological conditions, endothelial epsin expression increases, which causes acute inflammatory signaling to activate endothelial cells to express adhesion molecules on the cell surface. These proteins attract immune cells that exacerbate inflammation. In the long-term, epsins affect ER homeostasis. The two pathways merge to aggravate inflammation, causing more endothelial dysfunction to accelerate atherosclerosis.

## Data Availability

All data generated or analyzed supporting the findings of this study are available within the paper and its supplementary information files. All data are available from the corresponding author upon reasonable request.

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
