# Peer review of "Epsins Negatively Regulate Aortic Endothelial Cell Function by Augmenting Inflammatory Signaling"

_cells, 2021, doi:10.3390/cells10081918_

Round 1

Reviewer 1 Report

This is a manuscript from a group which marvelous expertise in epsin biology and vascular diseases. The group has published many excellent papers in the field to demonstrate a critical role of epsin in angiogenesis, inflammation and tumor growth. The current study is to expand its previous study on the molecular mechanism by which epsin regulates endothelial inflammation. With both in vitro and in vivo models, authors nicely demonstrated the role of epsin in regulating endothelial adhesion molecule expression and further explored that TLRs serve as underlying molecular targets for epsin-associated endothelial inflammation. This study indicates that endothelial epsin holds a promise as a target in the treatment of inflammatory vascular diseases. The research is well designed and conducted. The results are well presented with high quality images and clear figs. 

Author Response

We appreciate the positive comments from the reviewer!

Reviewer 2 Report

The manuscripts entitled “Epsins negatively regulate aortic endothelial cell function by augmenting inflammatory signaling” by Dong et al is very interesting and with great significance for the understanding of the mechanisms by which epsins regulate vascular functions. The findings are novel and provide new insights for the development of potential therapeutics for vascular diseases.

The major concern is the potential mistake of Figures 3G-J. Moma-2 may not be the right marker for PCR. Please check the accuracy of those figures, as well as the text description.

Minor suggestions:

  1. Although all the Western Blot results are very significant, it would be nice to present quantified data to support the images.
  2. For Fig 2, please explain the differences of ER stress markers in vivo and in vitro.
  3. For Figs 5E&G, please explain the normalization strategy.
  4. For Fig 6E, it would be nice to be presented similar to 6D.
  5. For Fig 7 summary figure, in addition to ox-LDL and 7-KC, will the authors consider to include TNFa and LPS, which were used in most of the experiment described in the manuscript.

Author Response

We greatly appreciate the comments from the reviewers. Below, we have addressed each concern from the initial round of review:

The major concern is the potential mistake of Figures 3G-J. Moma-2 may not be the right marker for PCR. Please check the accuracy of those figures, as well as the text description.

Reply: We apologize for this error in labelling. It should have been ‘E-selectin’ (not Moma-2), and we have corrected it. We also identified a mistake in the legend of Figure 3 (G to J), which we have corrected.

Minor suggestions:

  1. Although all the Western Blot results are very significant, it would be nice to present quantified data to support the images.

Reply: All blots are quantified and put in the supplement and text.

  1. For Fig 2, please explain the differences of ER stress markers in vivo and in vitro.

Reply: In theory, all ER stress markers can be used in vitro and in vitro; however, in practice, this is not true. So, we used antibodies known to work in each environment.

  1. For Figs 5E&G, please explain the normalization strategy.

Reply: We used “masked” area calculations for 3-NT or oxLDL antibody staining images. 

  1. For Fig 6E, it would be nice to be presented similar to 6D.

Reply: We have presented the data in a similar fashion.

  1. For Fig 7 summary figure, in addition to ox-LDL and 7-KC, will the authors consider to include TNFa and LPS, which were used in most of the experiment described in the manuscript.

Reply: We have included TNFα and LPS.